# Constitutional *BRCA1* and *MGMT* Methylation Are Significant Risk Factors for Triple-Negative Breast Cancer and High-Grade Serous Ovarian Cancer in Saudi Women

**DOI:** 10.3390/ijms25063108

**Published:** 2024-03-07

**Authors:** Nisreen Al-Moghrabi, Maram Al-Showimi, Amal Alqahtani, Osama Almalik, Hamed Alhusaini, Ghdah Almalki, Ajawhara Saad, Elaf Alsunayi

**Affiliations:** Cancer Epigenetics Section, Department of Molecular Oncology, King Faisal Specialist Hospital and Research Center, P.O. Box 3354, Riyadh 11211, Saudi Arabia; malshuwaymi@kfshrc.edu.sa (M.A.-S.); amalnasserq@gmail.com (A.A.); omalik@kfshrc.edu.sa (O.A.); hhusseini@kfshrc.edu.sa (H.A.); ghadaaliialmalki@gmail.com (G.A.); aljawhara1998@gmail.com (A.S.); efalsunie@gmail.com (E.A.)

**Keywords:** *BRCA1* methylation, *MGMT* methylation, breast cancer, TNBC, ovarian cancer, HGSOC

## Abstract

Breast cancer (BC) and ovarian cancer (OC) are rapidly increasing in Saudi Arabia. *BRCA1* and *MGMT* epimutations have been linked to a higher risk of these malignancies. The present research investigated the impact of these epimutations on the prevalence of BC and OC among Saudi women. DNA methylation was evaluated using methylation-specific PCR, whereas mRNA expression levels were assessed using qRT-PCR. We evaluated white blood cell (WBC)–*BRCA1* methylation in 1958 Saudi women (908 BC patients, 223 OC patients, and 827 controls). *MGMT* methylation was determined in 1534 of the 1958 women (700 BC patients, 223 OC patients, and 611 controls). *BRCA1* methylation was detected in 8.6% of the controls and 11% of the BC patients. This epimutation was linked to 13.8% of the early-onset BC patients (*p* = 0.003) and 20% of the triple-negative breast cancer (TNBC) patients (*p* = 0.0001). *BRCA1* methylation was also detected in 14% of the OC patients (*p* = 0.011), 19.4% of patients aged <55 years (*p* = 0.0007), and 23.4% of high-grade serous ovarian cancer (HGSOC) patients. In contrast, the *BRCA1* mutation was detected in 24% of the OC patients, 27.4% of patients aged ≥55 years, and 26.7% of the HGSOC patients. However, *MGMT* methylation was detected in 10% of the controls and 17.4% of the BC patients (*p* = 0.0003). This epimutation was linked to 26.4% of the late-onset BC patients (*p* = 0.0001) and 11% of the TNBC patients. *MGMT* methylation was also found in 15.2% of the OC patients (*p* = 0.034) and 19.1% of HGSOC patients (*p* = 0.054). Furthermore, 36% of the *BRCA1*-methylated patients and 34.5% of the *MGMT*-methylated patients had a family history of cancer, including breast and ovarian cancer. Notably, *BRCA1* and *MGMT* mRNA levels were greater in the WBC RNA of the BC patients and cancer-free methylation carriers than in that of the OC patients. Our data indicate that *BRCA1* and *MGMT* epimutations significantly contribute to the development of breast cancer and ovarian cancer in Saudi cancer patients. These blood-based biomarkers could help identify female patients at high risk of developing TNBC and HGSOC at an early age.

## 1. Introduction

An epimutation is a malfunction in the epigenetic regulatory mechanism that causes the aberrant repression of active genes and the reactivation of quiescent genes. This behavior has been identified as a potential cancer-susceptibility-enhancing mechanism [1]. DNA methylation, a well-studied epigenetic phenomenon, is widely recognized as a crucial gene-silencing mechanism in a variety of biological contexts. The procedure consists of adding a methyl group to the cytosine base within the CpG dinucleotide, resulting in the formation of 5-methylcytosine. Several cancer-related genes are inactivated by DNA promoter methylation in a variety of cancer types, resulting in genomic instability and assisting in the formation or progression of cancer. The presence of a methylated cancer-related gene in peripheral white blood cells (WBCs) is constitutive, implying that this epigenetic aberration exists in all animal tissues, thereby increasing the risk of developing cancer [2]. Such constitutional promoter methylation of key tumor suppressors is comparable to the inactivation of the same genes by germline mutations in the genesis of particular cancer types. This is exemplified by constitutional *BRCA1* promoter methylation, and numerous studies have demonstrated that this epimutation is substantially associated with early-onset triple-negative breast cancer (TNBC) subtypes and high-grade serous ovarian cancer (HGSOC) [3,4,5,6]. The error-free homologous recombination pathway [7], which repairs DNA double-strand breaks, is mediated by the BRCA1 protein. When the BRCA1 protein is absent or non-functioning due to mutations, cellular integrity is compromised, rendering cells more vulnerable to chromosomal rearrangements and mutations that have the potential to induce cancer. It is well established that germline *BRCA1* mutations are associated with familial breast and ovarian malignancies [8]. Those who have inherited *BRCA1* mutations have an increased risk of developing breast and ovarian cancer during their youth. In a similar vein, constitutive *BRCA1* methylation is a substantial risk factor for serous ovarian cancer and is associated with a 3.5-fold increased risk of early-onset breast cancer [3,9,10,11]. Notably, pathogenic germline *BRCA*1 mutations and methylated *BRCA1* promoter methylation have been shown to be mutually exclusive in breast and ovarian cancer [12]. Hence, there is ongoing research into constitutive *BRCA1* promoter methylation as a potential diagnostic biomarker in relation to the risk of developing breast and ovarian cancer.

O6-methylguanine DNA methyltransferase, or *MGMT*, is a DNA repair gene that removes alkyl groups from the O6 position of guanine nucleotides [13]. The inability to remove mutagenic adducts from guanine leads to DNA abnormalities and tumor growth when MGMT activity is lost [14,15]. *MGMT* is inactivated by DNA methylation in several human malignancies [16,17]. Prior studies have shown a correlation between *MGMT* promoter methylation and susceptibility to breast cancer [18], which is somewhat associated with older age [19]. In addition, there is evidence of a correlation between *MGMT* promoter methylation and clear cell and mucinous epithelial ovarian cancer subtypes [20]. We previously established that constitutional *MGMT* promoter methylation is highly related to both ovarian cancer and late-onset breast cancer in a study including 67 breast cancer patients and 82 ovarian cancer patients [4]. As a result, constitutive *MGMT* promoter methylation, like *BRCA1*, is a potential diagnostic biomarker for breast and ovarian cancer risk.

One of the most frequent malignancies among Arab women is breast cancer. Despite the fact that it is significantly lower than in many Western nations, there is mounting evidence that the incidence of breast cancer in Saudi Arabia is increasing rapidly [21]. Breast cancer incidences increased by 55% among Saudi women from 2001 to 2017, accounting for 30.9% of all cancer cases, with the median age of diagnosis rising to 51 years [22]. Ovarian cancer is the sixth most prevalent cancer among female patients, accounting for 3.3% of all female malignancies in Saudi Arabia. Ovarian carcinomas have a poor prognosis and a low overall five-year survival rate due to the absence of early-stage disease signs or symptoms. Thus, the discovery of non-invasive diagnostic biomarkers will have a direct impact on the early diagnosis and prevention of these cancers.

The objective of this study was to determine the frequencies of constitutional *BRCA1* promoter methylation and *MGMT* promoter methylation in Saudi women diagnosed with breast and ovarian malignancies, and to investigate potential associations with a family history of these illnesses. 

## 2. Results

### 2.1. Constitutional BRCA1 Promoter Methylation and MGMT Promoter Methylation Are Associated with BC in Saudi Breast Cancer Patients

In this study, we aimed to determine the roles of constitutional *BRCA1* promoter methylation and *MGMT* promoter methylation in the incidence of breast cancer in Saudi women. To this end, we screened a total of 1735 females for WBC *BRCA1* promoter methylation, 908 of whom had been diagnosed with breast cancer, and whose median age was 49 years, and 827 healthy female controls ranging in age from 15 to 50 years. The patient group included 492 individuals diagnosed with early-onset BC (<50 years old) and 416 patients diagnosed with late-onset BC (≥50 years old). Methylation-specific PCR was utilized to assess *BRCA1* promoter methylation in white blood cell DNA from both the control and BC groups. *BRCA1* methylation was detected in 71 (or 8.7%) of the 827 controls and 100 (or 11%) of the 908 BC patients; the mean age of these participants was 46.27 ± 11.67 (95%CI 43.96–48.57) years (Table 1A). This finding shows that there is no statistically significant relationship between constitutive *BRCA1* promoter methylation and overall breast cancer incidence (*p* = 0.104, OR = 1.32, 95% CI = 0.96 to 1.18). On the other hand, only 1311 individuals out of the 1735 females were screened for *MGMT* promoter methylation: 700 out of the 908 BC patients, and 611 out of the 827 controls. Out of the patient cohort, 370 were aged <50 years, while 330 were aged ≥50 years. *MGMT* methylation was detected in 61 (or 10%) of the 611 controls and in 17.4% (122 out of 700) of the BC patients; the mean age of these participants was 56 ± 12.46 (95% CI 53.89–58.36) years (Table 1B). Unlike the result for *BRCA1* promoter methylation, this result shows that there is a statistically significant relationship between constitutive *MGMT* promoter methylation and overall breast cancer incidence (*p* = 0.0003, OR = 1.9, 95% CI = 1.36 to 2.64), and there is a significant difference between the mean age of the two groups (Figure 1A).

### 2.2. BRCA1 Methylation Is Associated with Early-Onset BC, While MGMT Methylation Is Associated with Late-Onset BC

Taking patient ages into consideration, we discovered that of the 100 *BRCA1* methylation-positive patients, 68 were <50 years old and 32 were ≥50 years old, whereas of the 122 *MGMT* methylation-positive patients, 35 were <50 years old and 87 were ≥50 years old. These results showed that constitutional *BRCA1* methylation was found in 13.8% of early-onset BC patients (68 out of 492), compared to 7.7% of late-onset BC patients (32 out of 416). In contrast, constitutional *MGMT* methylation was found in 26.4% of late-onset patients (87 out of 330), compared to 9.5% of early-onset cases (35 out of 370). These results indicate that *BRCA1* methylation is linked to a higher risk of early-onset BC (*p* = 0.003, OR = 1.92, 95% CI = 1.24 to 2.99), while *MGMT* methylation is linked to a higher risk of late-onset BC (*p* = 0.0001, OR = 3.42, 95% CI = 2.23 to 5.24) compared to the control group (Table 1A,B).

### 2.3. Constitutional BRCA1 Promoter Methylation and MGMT Promoter Methylation Account for about One-Third of TNBC Instances in Saudi Breast Cancer Patients

In general, TNBC accounts for 10–20% of all BC cases [23]. The clinicopathological characteristics received from the pathology department for our sample of BC patients indicated that 15.8% of the patients (144/908) were of the TNBC subtype, with 29 instances being positive for *BRCA1* methylation. This finding suggests that the *BRCA1*-methylated group was more likely to exhibit TNBC subtypes, since 29% of the *BRCA1*-methylated patients (29/100) had TNBC compared to 14.2% of the unmethylated cases (115/808). When compared to the controls, this result indicates a substantial relationship between *BRCA1* methylation and TNBC in Saudi BC patients (*p* = 0.0001 with an OR of 2.68 and a 95% CI of 1.64 and 4.31) (Table 1A). Similarly, of the 700 BC patients examined for *MGMT* promoter methylation, 110 (15.7%) had TNBC, with 12 having *MGMT* methylation. This result indicates that, unlike *BRCA1* methylation, constitutional *MGMT* methylation does not correlate with incidents of TNBC, as only 10.1% of the *MGMT* methylated cases (12/122) were of the TNBC subtype compared to 16.8% of the unmethylated cases (98/578) (*p* = 0.153 with an OR of 1.53 and a 95% CI of 0.85 and 2.77) (Table 1B). Nevertheless, our data indicate that about 30% of TNBC among Saudi BC patients can be attributed to *BRCA1* (29/144, 20%) and *MGMT* (12/110, 11%) epimutations.

### 2.4. Constitutional BRCA1 Promoter Methylation and MGMT Promoter Methylation Contribute to a Greater Proportion of OC in Saudi Women Than Mutant BRCA1

Additionally, we aimed to determine the roles of constitutional *BRCA1* promoter methylation and *MGMT* promoter methylation in the incidence of ovarian cancer in Saudi women. To this end, we recruited 223 women with ovarian cancer whose median age was 57 years and who had been diagnosed in the oncology department of King Faisal Specialist Hospital and Research Centre from 2017 to 2023. The hospital gave information related to germline *BRCA1* mutations for 167 patients. As ovarian cancer is often diagnosed between the ages of 55 and 64, a cut-off of 55 years was selected [24]. The patient group included 103 women aged <55 years old and 120 aged ≥55 years old. *BRCA1* methylation was detected in 14% of the patients (32/223), whose mean age was 53.47 ± 13.10 (95% CI 48.75–58.19), and *MGMT* methylation was detected in 34 patients (15%), whose mean age was 54.61 ± 15.62 (95% CI 49.07–60.15). These findings indicate a statistically significant association between *BRCA*1 and *MGMT* epimutations and the total incidence of OC in Saudi females (*p* = 0.01, OR = 3.25, 95% CI = 1.58–2.78, Table 2A, and *p* = 0.034, OR = 1.62, 95% CI = 1.03–2.54, Table 2B, respectively). Taking patient ages into consideration, we found that 19.4% (20/103) of the *BRCA*1-methylation-positive patients were <55 years old, and 10% (12/120) were ≥55 years old. In comparison to the controls, these data indicate that constitutional *BRCA1* promoter methylation is statistically associated with an elevated risk of OC in people under the age of 55 (*p* = 0.0007, OR = 2.56, 95% CI = 1.48 to 4.4) (Table 2A). However, there was no correlation between constitutional *MGMT* promoter methylation and patient age, as 15.5% of the *MGMT*-methylation-positive patients (16/103) were <55 years old and 14% (17/120) were ≥ 55 years old (Table 2B). Altogether, our findings show that *BRCA1* and *MGMT* epimutations account for 29.5% of OC in Saudi patients. Of the patients with *BRCA1* mutations, on the other hand, 24% (40/167) were positive for mutated *BRCA1*, and all of them were negative for *BRCA*1 and *MGMT* epimutations. The mean age of these patients was 56.80 ± 11.40 (95% CI 52.15–60.45). Of these patients, 19.4% (14/72) were <55 years old and 27.4% (26/95) were ≥55 years old (Table 2C). Unlike the breast cancer patients, there was no significant difference between the mean ages of the patients in these three groups (Figure 1B).

### 2.5. Constitutional BRCA1 Promoter Methylation and MGMT Promoter Methylation Account for a Higher Proportion of HGSOC in Saudi Women with Ovarian Cancer Than Mutant BRCA1

In our OC cohort, the clinicopathological parameters obtained from the pathology department revealed that 47 patients were of the HGSOC type, among whom we identified 11 cases positive for *BRCA1* methylation and 9 cases positive for *MGMT* methylation. These results reveal that 42.5% of the HGSOC—23.4% (11/47) for *BRCA1* and 19.1% (9/47) for *MGMT*—can be attributed to *BRCA1* and *MGMT* epimutations. However, for the group tested for the germline *BRCA1* mutation, 26.6% (12/45) had HGSOC (Table 2A–C). Compared to the controls, our findings indicate a highly significant association between *BRCA1* methylation and HGSOC (*p* = 0.001 OR = 3.25, 95% CI = 1.58–6.68) (Table 2A), but only a marginally significant association between *MGMT* methylation and HGSOC (*p* = 0.054, OR = 2.13, 95% CI = 0.98–4.62) (Table 2B).

### 2.6. Patients with BRCA1- and MGMT-Methylated Cancers Have a Family History of Cancer

In a previous study, we examined the possibility of constitutional *BRCA1* and *MGMT* promoter methylation being passed down from mother to daughter [4]. Here, we sought to see whether there was a link between *BRCA1*- and *MGMT*-methylated breast and ovarian cancer and each patient’s family history of cancer. Out of the 132 *BRCA1*-methylation-positive breast and ovarian cancer cases, the family history was known for only 75 cases. Notably, 36% (27/75) of the cases had a family history of cancer, of which 63% (17/27) had breast and ovarian cancer and 37% (10/27) had other types of cancer (Table 3A,B). For *MGMT*-methylated breast and ovarian cancer patients, a family history of cancer was determined in 113 patients out of the 156 *MGMT*-positive cases. Notably, 34.5% (39/113) had a family history of cancer, of which 41% (16/39) had breast, uterine, and endometrial cancers, and 59% (23/39) had other types of cancer, among which 26.1% (6/23) had colon cancer (Table 4A,B). Overall, our findings suggest a possible link between *BRCA1* and *MGMT* epimutations and the occurrence of cancer in the family.

### 2.7. BRCA1- and MGMT-Methylated Breast Cancer Patients, as Well as Cancer-Free Methylation Carriers, Express High Levels of BRCA1 and MGMT mRNA

To see whether there is a link between constitutional promoter methylation and gene expression in WBCs, we assessed the mRNA levels of methylated *BRCA1*- and *MGMT*-methylated genes in WBC RNA from breast and ovarian cancer patients, as well as that of cancer-free (CF) methylation carriers, and compared them with those of the controls using quantitative real-time PCR. Notably, we identified no change in *BRCA1* and *MGMT* mRNA levels in the ovarian cancer patients (Figure 2A,C). However, the expression of both genes was greater in the breast cancer patients (*p* = 0.033 for *BRCA1* and 0.049 for *MGMT*) (Figure 2B,D). Notably, the CF methylation carriers expressed significantly higher levels of the two genes (*p* = 0.0001 for *BRCA1* and 0.015 for *MGMT*) than the breast cancer patients (Figure 2E,F).

## 3. Discussion

The discovery of minimally invasive biomarkers for the identification of asymptomatic cancer is of the utmost importance to enhance early cancer risk prediction for prevention and early diagnosis. Constitutional *BRCA1* promoter methylation and *MGMT* promoter methylation have been shown to be associated with an increased risk of ovarian cancer and breast cancer [3,4,25]. In this study, we assessed the contribution of *BRCA1* and *MGMT* epimutations to breast and ovarian cancer in women from Saudi Arabia. 

We conducted a comprehensive analysis of a group of 1958 Saudi women, comprising breast and ovarian cancer patients and controls, to determine the prevalence of *BRCA1* and *MGMT* epimutations. Although *BRCA1* epimutation was not shown to be significantly associated with BC, it was found to be more common than the germline *BRCA1* mutation (11% vs. 8.3%, respectively) [26]. However, both forms of *BRCA1* gene abnormalities are strongly linked to the development of BC before the age of 50 [22,27,28]. *MGMT* methylation, on the other hand, was found to be strongly linked to BC and, in particular, to late-onset BC, and this finding is in concordance with our previous study [4]. Given the higher frequency of TNBC among younger females [29,30,31], it is not surprising that our findings demonstrate a strong correlation between TNBC and constitutional *BRCA1* methylation rather than constitutional *MGMT* methylation. Nevertheless, these data reveal that *BRCA1* and *MGMT* epimutations account for about 28% of overall BC and 39% of the TNBC subtype among Saudi female BC patients.

As with pathogenic *BRCA1* mutations [26], *BRCA1* epimutation is more prevalent in OC patients than in BC patients, especially among those <55 years old (19.4% vs. 13.8%, respectively). This finding is consistent with our prior work, which observed that the incidence of *BRCA1* epimutation in OC patients was twice as high as in BC patients. In contrast to BC, the prevalence of pathogenic *BRCA1* mutations in OC patients surpasses that of *BRCA1* epimutation, particularly in patients aged ≥55 (27.4% vs. 10%, respectively). On the other hand, *MGMT* epimutation exhibits a higher incidence rate in BC patients than in OC patients, in particular among the elderly (26.4% vs. 14%, respectively). Notably, our findings show that *BRCA1* and *MGMT* epimutations account for almost 30% of all OC and almost 43% of severe HGSOC in Saudi patients. Given that the prevalence of pathogenic *BRCA1* mutations varies between 24% and 41% [26,30], this suggests that potentially up to 70% of ovarian cancer cases might be anticipated and avoided at an early stage. 

One-third of the *BRCA1*- and *MGMT*-methylation-positive patients had a strong family history of cancer, including breast, ovarian, and colon cancers. These data are consistent with our previous study, which found that 77% of CF *BRCA1* epimutation carriers had a family history of cancer, with 70% of instances being breast and ovarian cancer [25]. While we did not have access to blood samples to determine the methylation status of patients’ relatives in the present study, the data suggest that epimutation carriers originate from families with epigenetic abnormalities. Indeed, in a previous study using 290 mother–newborn pairs, we reported that 20% of the mothers carrying *BRCA1* epimutations and 31% of the mothers carrying *MGMT* epimutations gave birth to *BRCA1* and *MGMT* carriers, respectively. Furthermore, *BRCA1* mother carriers delivered *MGMT* newborn carriers, and vice versa. 

Based on our data, almost 19% of CF Saudi women are carriers of constitutional epimutations: 10% carry *MGMT* epimutations and 8.6% carry *BRCA1* epimutations from early on in life. Mounting evidence substantiates the notion that such persons are at a heightened risk of developing breast and/or ovarian cancer [4,5,9,25,31,32,33]. We previously demonstrated that the levels and arrangements of CpG island methylation at the *BRCA1* promoters in WBCs are comparable among cancer patients, newborns, and adult *BRCA1* methylation carriers [4]. In addition, it has been shown that there is a concordance between *BRCA1* epimutations in WBCs and tumor tissues [25]. These findings raise the possibility that methylation takes place as a single-cell occurrence during early embryonic development, and that it is followed by clonal expansion across all germ layers [33]. Moreover, studying the molecular effects of *BRCA1* epimutation in WBCs has shown that there are cancer-related molecular changes that can occur in adult carriers and, more importantly, in newborn carriers. These changes are similar to those seen in females who have been diagnosed with breast cancer and ovarian cancer [5,9,25]. Thirdly, in a recent study [5], we found that the WBCs of CF *BRCA1* methylation carriers had less ILR2G (a T cell functional molecule), suggesting that these carriers have reduced antitumor immunity. This discovery aligns with a recent investigation that suggested that modified *BRCA1* expression in peripheral T cells could lead to aberrant transcription, which is linked to antitumor immunity, and which could potentially contribute to the elevated cancer risk observed in women carrying *BRCA1* mutations [34]. It is possible that a similar mechanism is at play in *BRCA1* methylation carriers. Finally, our new data show that the amounts of *BRCA1* and *MGMT* mRNA in WBC RNA are much higher in patients with breast cancer and CF epimutation carriers compared to the control group. These findings contrast with those for tumor tissues, where changes in the methylation status of the *BRCA1* and *MGMT* promoters alter the degree of mRNA expression [35,36,37,38,39]. While the underlying biology is unclear, this result indicates that CF epimutation carriers have molecular aberrations similar to those seen in cancer patients. Collectively, these data evidence the increased risk of cancer development among epimutation carriers, indicating that these epimutations could be used as possible risk factors for early cancer prediction. 

In conclusion, our findings show that constitutional *BRCA1* and *MGMT* methylation play an important role in the development of breast and ovarian cancer in Saudi female patients; the results are summarized in Figure 3. The fact that these epimutations appear at a young age enables the identification of young women who are more prone to developing HGSOC or TNBC. The reversibility of DNA methylation presents the potential for cancer prevention. Therefore, if these methylations are identified at an early stage, it is feasible that their effects can be mitigated using non-invasive interventions like dietary supplements. Curcumin, a substance with demethylating capabilities [40] and the capacity to improve antitumor immunity [41], has promise as a cancer preventative agent, particularly for *BRCA1*-methylation carriers. This merits more investigation in future studies.

## 4. Materials and Methods

### 4.1. Study Population 

A total of 1958 females were included in this study, 908 of whom had been diagnosed with breast cancer, 223 of whom has been diagnosed with ovarian cancer, and 827 of whom were CF volunteers. The blood samples, amounting to 10 mL each, were collected from the patients during their visits to the Department of Oncology at King Faisal Specialist Hospital and Research Centre in Riyadh, Saudi Arabia, from November 2017 to June 2023. The Department of Oncology gave information regarding the patients’ age, family history, and germline *BRCA1* mutation status. The age distribution of the breast cancer patients varied between 20 and 94 years, with a median age of 49 years, and the age distribution of the ovarian cancer patients varied between 19 and 88 years, with a median age of 55 years. The Department of Pathology gave information regarding the patients’ histological grade, estrogen receptor status, and progesterone receptor status. The CF volunteers were between 15 and 50 years old. Ethical permission was obtained (approval no. RAC #2170017) from the Human Research Ethics Committee of King Faisal Specialist Hospital and Research Centre. Written informed consent was obtained from all participants.

### 4.2. DNA and RNA Isolation from WBCs

BD Vacutainer EDTA blood collection tubes (Becton, Dickinson and Company, Franklin Lakes, NJ, USA) were used to collect each blood sample. The tubes were immediately centrifuged at 2000× *g* for 10 min at 4 °C. The WBC layer was separated in equal parts and transferred to two 2 mL Eppendorf tubes. The first contained 900 mL of RBC lysis solution for DNA extraction with the Gentra Puregene Blood Kit (Qiagen GmbH, Hilden, Germany), and the second contained 1.2 mL of RNALater solution for RNA extraction with the RiboPure Blood Kit (Ambion; Thermo Fisher Scientific, Inc. Waltham, MA, USA).

### 4.3. Methylation-Specific Polymerase Chain Reaction

A total of 2 µg of WBC DNA was treated with sodium bisulfate before being purified using the EpiTect Bisulfite Kit (Qiagen GmbH) according to the manufacturer’s instructions. *BRCA1* and *MGMT* PCR primers that differentiate between methylated and unmethylated DNA were used to amplify the treated DNA (Table 5) [4]. The PCR was carried out in a Veriti Thermal Cycler (Applied Biosystems, Foster City, CA, USA). For the methylated *BRCA1* primers, an initial cycle at 95 °C for 1 min was followed by 40 cycles at 65 °C for 30 s, 72 °C for 30 s, and a final extension at 72 °C for 7 min. For the methylated *MGMT* primers, an initial cycle at 95 °C for 1 min was followed by 40 cycles at 59 °C for 30 s, 72 °C for 30 s, and a final extension at 72 °C for 7 min. The PCR products were electrophoresed on 2% agarose gels and stained with ethidium bromide. The Molecular Imager Gel Doc XR System was used to visualize the bands. Totally methylated bisulfite-treated DNA was used as a positive control. Every reaction was performed at least twice.

### 4.4. Reverse Transcription Quantitative PCR (RT qPCR) 

In a 20 µL reaction, 1 µg of pure RNA was reverse-transcribed to single-stranded cDNA using Superscript III, reverse transcriptase, and random hexamers (Applied Biosystems; Thermo Fisher Scientific, Inc.; cat. no. 4368814). qPCRs with specific primers for the *MGMT* and *BRCA1* transcripts were conducted using actin beta (*ACTB*) as a housekeeping gene (Table 5). A CFX96 Real-Time System (Bio-Rad Laboratories, Inc., Hercules, CA, USA) was used for the PCR, which used SYBR Green (RT2 SYBR Green Fluor qPCR Master mix; cat. no. 330513; Qiagen GmbH). The qPCR thermal cycling settings were as follows: For *BRCA1,* an initial cycle at 95 °C for 30 s was followed by 44 cycles at 95 °C for 15 s, and 59 °C for 30 s. For *MGMT,* an initial cycle at 95 °C for 30 s was followed by 44 cycles at 95 °C for 15 s, and 60 °C for 30 s. The relative *MGMT* and *BRCA1* expressions were calculated using the 2^−ddCt^ method [42]. For patients with breast cancer and ovarian cancer, as well as CF female carriers, the fold changes in mRNA expression was compared to those of the unmethylated CF females. 

### 4.5. Statistical Analysis

Fisher’s exact test was performed to assess the associations between *BRCA1* and *MGMT* promoter methylation, age, and clinicopathological features of BC and OC. An unpaired *t* test was performed to determine the statistical significance of gene expression in the different groups (breast cancer vs. controls, ovarian cancer vs. controls, and CF carriers vs. controls). A one-way ANOVA with Dunnett’s multiple comparison test was performed to compare multiple groups. GraphPad version 9.1.0 (GraphPad Software, Inc., La Jolla, CA, USA) was used for all analyses, and *p* < 0.05 was used to indicate a statistically significant difference.

## Figures and Tables

**Figure 1 ijms-25-03108-f001:**
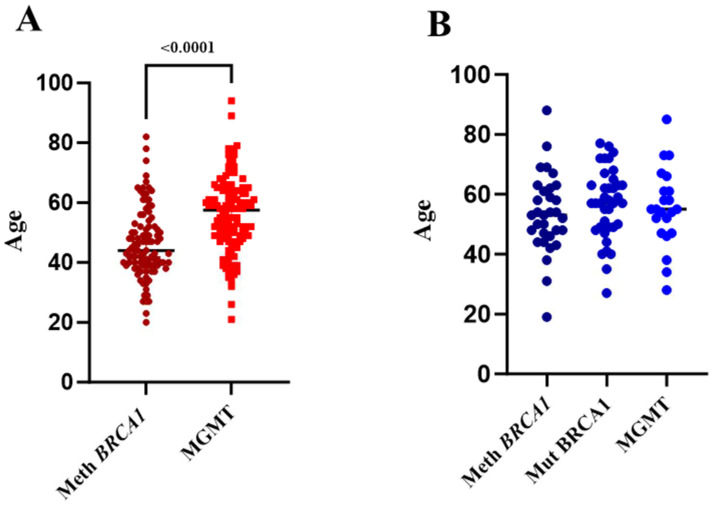
The mean age of cancer patients with constitutional *BRCA1* and *MGMT* methylation. Methylation-specific PCR was used to determine *BRCA1* and *MGMT* promoter methylation in white blood cell DNA from the BC and OC patients. (**A**) Comparison between the mean ages of the *BRCA1*- and *MGMT*-methylated BC-positive patients. (**B**) Comparison between the mean ages of the *BRCA1*-methylated, *BRCA1*-mutated, and *MGMT*-methylated OC-positive patients. Meth—methylated, Mut—mutated.

**Figure 2 ijms-25-03108-f002:**
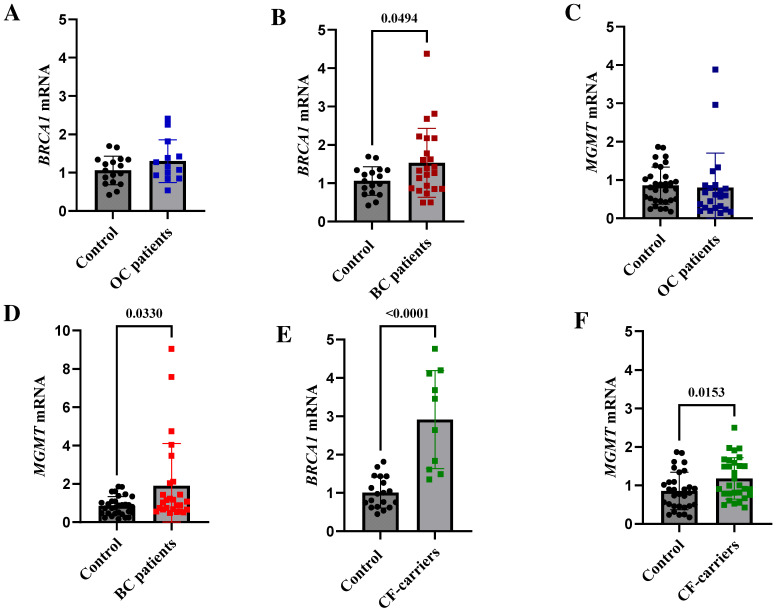
Expression of *BRCA1* and *MGMT* mRNA in peripheral WBCs in epimutation-positive cancer patients. The expression of mRNA was measured using RT-qPCR. (**A**,**B**) Analysis of *BRCA1* mRNA expression in WBCs of OC and BC patients with methylated *BRCA1*. (**C**,**D**) Analysis of *MGMT* mRNA expression in WBCs of OC and BC patients with methylated *MGMT*. (**E**,**F**) Analysis of *BRCA1* and *MGMT* mRNA expression in WBCs of cancer-free (CF) *BRCA1* and *MGMT* methylation carriers, respectively. Error bars represent the mean ± SD. RT-qPCR—reverse transcription-quantitative polymerase chain reaction, WBC—white blood cells, BC—breast cancer, OC—ovarian cancer.

**Figure 3 ijms-25-03108-f003:**
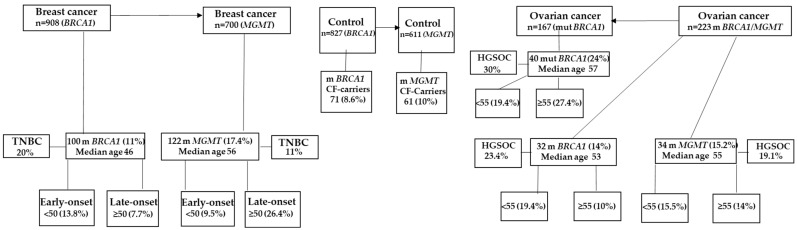
A schematic diagram summarizing the results. TNBC—triple negative breast cancer, HGSOC—high-grade serous ovarian cancer, m *BRCA1*—methylated *BRCA1*, m *MGMT*—methylated *MGMT*, mut *BRCA1*—mutated *BRCA1*, CF—cancer free.

**Table 1 ijms-25-03108-t001:** (**A**) Association of constitutional *BRCA1* promoter methylation with TNBC and early-onset BC. (**B**) Association of constitutional *MGMT* promoter methylation with late-onset BC.

	Meth (n%)	Unmeth (n%)	*p* Value, OR (95% CI)
(**A**)			
**Control** (n = 827)	71 (8.6)	756 (91,4)	
**BC** (n = 908)	100 (11)	808 (89)	0.104, 1.32 (0.96, 1.18)
**Age**			
**<50** n = 492	68 (13.8)	424 (86.2)	0.003, 1.92 (1.24, 2.99)
**≥50** n = 416	32 (7.7)	384 (92.3)	
**TNBC** n = 144	29 (20)	115 (80)	0.0001, 2.68 (1.64, 4.31)
(**B**)			
**Control** (n = 611)	61 (10)	550 (90)	
**BC** (n = 700)	122 (17.4)	578 (82.6)	0.0003, 1.9 (1.36, 2.64)
**Age**			
**<50** n = 37	35 (9.5)	335 (90.5)	
**≥50** n = 330	87 (26.4)	243 (73.6)	0.0001, 3.42 (2.23, 5.24)
**TNBC** n = 110	12 (11)	98 (89)	0.153, 1.53 (0.85, 2.77)

BC—breast cancer, Meth—methylated, Unmeth—unmethylated, TNBC—triple-negative breast cancer.

**Table 2 ijms-25-03108-t002:** (**A**) Association of constitutional *BRCA1* promoter methylation with HGSOC. (**B**) Association of constitutional *MGMT* promoter methylation with HGSOC. (**C**) Association of mutated *BRCA1* with ovarian cancer.

	Meth (n%)	Unmeth (n%)	*p* Value, OR (95% CI)
(**A**)			
**Control** (n = 827)	71 (8.6)	756 (91.4)	
**OC** (n = 223)	32 (14)	191 (86)	0.011, 1.78 (1.14, 2.78)
**Age**			
** <55** n = 103	20 (19.4)	83 (80.6)	0.0007, 2.56 (1.48, 4.4)
**≥55** n = 120	12 (10)	108 (90)	
**HGSOC** n = 47	11 (23.4)	36	0.001, 3.25 (1.58, 6.68)
(**B**)			
**Control** (n = 611)	61 (10)	550 (90)	
**OC** (n = 223)	34 (15.2)	189 (84.8)	0.034, 1.62 (1.03, 2.54)
**Age**			
**<55** n = 103	16 (15.5)	87 (84.5)	
**≥55** n = 120	17 (14)	103 (86)	
**HGSOC** (n = 47)	9 (19.1)	38	0.054, 2.13 (0.98, 4.62)
(**C**)			
	**Mut (n%)**	**WT (n%)**	***p* Value, OR (95% CI)**
**OC** (n = 167)	40 (24)	127 (76)	
**Age**			
**<55** n = 72	14 (19.4)	58 (80.5)	0.236, 1.56 (0.74, 3.26)
**≥55** n = 95	26 (27.4)	69 (27.6)	
**HGSOC** (n = 45)	12 (26.7%)	33	

OC—ovarian cancer, HGSOC—high-grade serous ovarian cancer, Meth—methylated, Unmeth—unmethylated, Mut—mutated, WT—wild type.

**Table 3 ijms-25-03108-t003:** (**A**) Family history of cancer in WBC *BRCA1*-methylated BC patients. (**B**) Family history of cancer in WBC *BRCA1*-methylated OC patients.

Sample #	Age	Affected FM	Type of Cancer
(**A**)			
162	42	Grandmother	BC at age 70
165	50	Sister	BC
199	52	Mother	BC
235	31	Cousin	BC
237	48	Cousin	BC
315	64	Sister	BC
329	59	Mother, Sister, Aunt	BC
409	65	Sister	BC
547	59	Sister	BC
573	20	ND	FH of BC
617	39	Sister	BC
642	29	Cousin	BC
390	55	Mother	OC
172	46	Mother	Jaw cancer
181	34	Mother	Thyroid cancer
275	61	ND	Bone and Lung cancer
429	65	Mother	Oropharyngeal cancer
587	44	Father	Urinary bladder cancer
605	63	Sister	ND
650	74	Daughter	Colon cancer
(**B**)			
60	61	2 cousins	Breast and Uterine cancer
123	62	Sister	BC
104	50	2 cousins	BC
30	46	Sister	Cervical cancer
31	53	Aunt	ND
122	63	Son	Benign tumor in neck
132	69	Father	Brain cancer

BC—breast cancer, OC—ovarian cancer, FM—family member, FH—family history, ND—not determined.

**Table 4 ijms-25-03108-t004:** (**A**) Family history of cancer in WBC *MGMT*-methylated BC patients. (**B**) Family history of cancer in WBC *MGMT*-methylated OC patients.

Sample #	Age	Affected FM	Type of Cancer
(**A**)			
136	51	FH in 2 of 2nd generation	BC
212	69	Cousin	BC
341	60	Sister	BC
352	58	Mother	BC at old age
514	60	Sister	BC
638	66	Daughter	BC
566	39	Grandma	BC
646	64	FH	BC
559	76	Cousin	BC
625	41	Mother	BC
689	60	Aunt	BC at old age
		Uncle	Pancreatic cancer
702	76	Mother	BC
		Cousin	Oropharyngeal cancer
712	49	Mother	BC at 70 years old
733	78	Cousin	BC
353	52	Sister	Uterine cancer
711	65	Sister	Endometrial cancer
		Father	Neck and prostate cancer
28	47	Father	ND
39	62	FH	Liver cancer
		Cousin	Colon cancer
167	52	Uncle	Thyroid cancer
200	59	Mother	Bowel cancer
428	51	Mother	Thyroid cancer
464	68	Brother	Prostate cancer
465	53	Brother	Bladder cancer
511	59	Sister	Bone marrow cancer
522	67	Daughter	Spinal cancer
		Brother	Liver and prostate cancer
556	58	Uncle	Colon cancer
562	63	Cousin	Abdominal cancer
569	35	Father	Renal cancer
		Uncle	Bladder cancer
616	89	Mother	Pancreatic cancer
619	65	Brother	Colon cancer
659	50	Mother	Oral cavity cancer
(**B**)			
53	73	Mother	Uterine cancer
55	55	Son	Thyroid cancer
59	52	Sister	Uterine cancer
79	64	Sister (1)	Endometrial and thyroid cancer
		Sister (2)	Colon Cancer
163	61	Two brothers	Colon cancer

BC—breast cancer, FM—family member, ND—not determined, FH—family history, OC—ovarian cancer.

**Table 5 ijms-25-03108-t005:** RT-quantitative PCR and MSPCR primers.

Primer Name	Primer Sequence	Annealing Temp
RT *BRCA1*	F5′-TGTAGGCTCCTTTTGGTTATATCATTC-3′ R5′-CATGCTGAAACTTCTCAACCAGAA-3′	59 °C
β-Actin	F5′-TCCCTGGAGAAGAGCTACGA-3′ R5′-TGAAGGTAGTTTCGTGGATGC-3′	59 °C
RT *MGMT*	F5′-GCGTTCGACGTTCGTAGGT-3′ R5′-CACTCTTCCGAAAACGAACG-3′	60 °C
	F5′-AAACTGGAACGGTGAAGG TG-3′	
β-Actin	R5′-AGTGGGGTGGCTTTTAGGAT-3′	60 °C
M *BRCA1*	F5′-GGTTAATTTAGAGTTTCGAGAGACG-3′	
	R5′-TCAACGAACTCACGCCGCGCAATCG-3′	65 °C
UM *BRCA1*	F5′-GGTTAATTTAGAGTTTTGAGAGATG-3′	
	R5′-TCAACAAACTCACACCACACAATCA-3′	65 °C
M *MGMT*	F5′-TTTCGACGTTCGTAGGTTTTCGC-3′	
	R5′-GCACTCTTCCGAAAACGAAACG-3′	59 °C
UM *MGMT*	F5′-TTTGTGTTTTGATGTTTGTAGGTTTTTGT-3′	
	R5′-AACTCCACACTCTTCCAAAAACAAAACA-3′	59 °C

M—methylated, UM—unmethylated.

## Data Availability

All data generated or analyzed during this study are available from the corresponding author on reasonable request. The data are not publicly available due to Confidentiality and privacy.

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
