# Peer review of "Constitutional BRCA1 and MGMT Methylation Are Significant Risk Factors for Triple-Negative Breast Cancer and High-Grade Serous Ovarian Cancer in Saudi Women"

_ijms, 2024, doi:10.3390/ijms25063108_

Round 1

Reviewer 1 Report

Comments and Suggestions for Authors

In the submitted manuscript authors analyzed constitutional methylation status of BRCA1 and MGMT genes in breast cancer, ovarian cancer patients and healthy controls, and associated these with age, cancer subtype, and BRCA1 and MGMT expression levels.

Due to the large sample size, obtained results are robust, and manuscript is quite well written. However, there are several drawbacks which must be corrected or further improved:

1) Lines 55-57: To be completely precise, HR is mediated by BRCA1 protein, not gene, and "cellular integrity is compromised" not only when the BRCA1 protein is missing, but also when the BRCA1 protein IS present, but is non-functional (due to mutations)!

2) Lines 66 and 79: Predictive biomarkers are those which predict therapy efficiency, while in your case it is about potential diagnostic biomarkers!

3) Sentence in lines 87-88 must be referenced.

4) Authors should recheck text that BRCA1 and MGMT are always italicized when they are in the context as genes or mRNA.

5) All abbreviations, even BC and OC, presented in tables must be explained in tables' footnotes, wile in right columns of all tables also percentages should be presented.

6) Figure 1 is too blurry, panels' titles should be removed, and Y-axis properly named as just "Age", since only those horizontal lines present mean age!

7) Since selecting an age cut-off of 50 years for breast cancer was referenced in 'Discussion', it must be explained in the text also why age cut-off of 55 years was used for ovarian cancer.

8) All p-values higher than 0.001 should uniformly be presented with only three decimals, while results of statistical analyses are missing from Table 2C.

9) Figure 2 is also too blurry, panels' titles should be removed, Y-title precisely named (mentioning a gene), as well as bars, since from figure it is unclear what "Cont" presents. Panels should be properly consecutively named A to F, mentioned such in figure legend, as well as cited in the text.

10) Since Table 5 in present form provides already presented data, it is actually redundant and should be omitted, except the summary data is presented is some other form.

11) It must be statistically proven that the age of those three groups of patients and controls does not statistically significantly differ among groups (lines 481-482).

12) It is quite strange that something must be centrifuged at precisely 1962 x g (line 488).

13) Line 491 and elsewhere: vendor of all commercial kits and chemicals must be provided.

14) For MS-PCR, used PCR machine must be stated, as well as precisely described all other methodological steps after PCR. It must be also explained in the manuscript what is considered as methylated and unmethlythed BRCA1/MGMT, i.e., how many CpGs were analyzed!

15) Line 56: Proper name of the gene is actin beta and its symbol is ACTB.

16) Proper formula is 2-ddCt and reference DOI: 10.1006/meth.2001.1262 should be cited.

Comments on the Quality of English Language

1) It is unclear what means "TNG" in line 165.

2) Proper sentence in lines 180-183 would be "To this end, we recruited women with ovarian cancer with a median age of 57 years, diagnosed in the oncology department of King Faisal Specialist Hospital and Research Centre from 2017 to 2023."

3) Proper therm is "unmethylated", not "un meth..." or "un-meth...".

4) Lines 419-420: It is unclear what actually means "we identified aberrant transcription in T cell functional molecules in WBC".

5) Line 519: Since correlation coefficient has not been calculated, phrase "correlations" should be avoid and rather "associations" used.

6) Lines 523-524: It is unclear what means "different groups" in the sentence "A one‑way ANOVA with Dunnett's multiple comparison tests was performed to compare the different groups.".

Reviewer 2 Report

Comments and Suggestions for Authors

This research manuscript explores the correlation between constitutional BRCA1 and MGMT methylation and the prevalence of breast cancer and ovarian cancer among Saudi women. The study employs methodologies, including methylation-specific PCR and qRT-PCR, to evaluate DNA methylation and mRNA expression levels.

Comments and Questions for Improvement:

Comment 1: The author should provide and confirm the results with additional methods, such as DNA Methylation Sequencing, to bolster the findings and strengthen the support for the hypothesis.

Comment 2: The clarity of the collection of blood samples is essential. It's unclear whether the samples were obtained before treatment initiation or if they include patients undergoing treatment. Clarification on this point would enhance the study's transparency and validity.

Comment 3: Given the specific focus on Saudi women, it would be valuable to discuss the generalizability of these findings to other populations. Are there unique genetic or environmental factors in Saudi Arabia that might influence the association between epimutations and cancer risk?

Reviewer 3 Report

Comments and Suggestions for Authors

Al-Moghrabi et al. have aimed to determine the frequency of constitutional BRCA1 and MGMT promoter methylation in Saudi women diagnosed with breast and ovarian malignancies as well as to investigate potential associations with a family history of these illnesses. It is an interesting piece of work executed and written well. However, the following points need to be addressed for a possible publication in IJMS.

1.    The abstract is well written with clarity and sufficient quantitative results.

2.    Two more keywords such as “breast cancer” and “ovarian cancer” should be added.

3.    Both the captions and table contents of individual parts (A & B or A, B & C) of each table should be combined as one single table to have a more comprehensive data observation. To this end, all the tables should be reformatted.

4.    In Tables 3 and 4, “F M” and “F H” should be corrected as “FM” and “FH”, respectively.

5.    All the abbreviations used in both Tables and Figures should be provided in full form in the respective Table footnotes and Figure captions.

6.    A recently published article featuring the inhibition effect of breast cancer cells and tumors in mice by carotenoid extract and nanoemulsion prepared from sweet potato peel should be cited (https://doi.org/10.3390/pharmaceutics14050980).

7.    A schematic diagram showing the workflow, methodology and key outcomes should be provided at the end of discussion (before the conclusion), which gives easy take-home points for the readers.

8.    All the purchase details of chemicals/reagents and instruments/equipment/software/kits should be provided as state, city, and country in the case of USA as well as city and country in the case of other countries. Also, the authors can just mention the company name for the second instance.

Comments on the Quality of English Language

Minor editing of English language required

Round 2

Reviewer 1 Report

Comments and Suggestions for Authors

Authors have properly addressed to all my concerns, except that we obviously did not understand each other regarding the suggestion 6) "in right columns of all tables also percentages should be presented" by which I did not mean that % sign should be put next to percentage, but that except just a number of samples (n), also percentage (%) must be provided in the third column of Tables 1A,B and Tables 2A,B,C. For example, in the second column of Table 2C there should be: OC (n= 167) 40 (24.0) [for Mut (n/%)] 127 (76.0) [for WT (n/%)]!

Author Response

Comment: The suggestion 6) "in right columns of all tables also percentages should be presented"

Response: Changes have been made. 

I wish I understood the intended message better.

Reviewer 2 Report

Comments and Suggestions for Authors

I find the authors' responses to be comprehensive and convincing, and I agree to accept the revised manuscript. Congratulations to the authors!

Author Response

Comments and Suggestions for Authors

I find the authors' responses to be comprehensive and convincing, and I agree to accept the revised manuscript. Congratulations to the authors!

Response: Thanks.